# Epidemiological study of cervical cord compression and its clinical symptoms in community-dwelling residents

**Toru Hirai[ID], Koji Otani\*, Miho Sekiguchi, Shin-ichi Kikuchi, Shin-ichi Konno**

Department of Orthopaedic Surgery, Fukushima Medical University School of Medicine, Fukushima, Japan

\* kotani@fmu.ac.jp

**Data Availability Statement:** The underlying data was obtained from the collaboration with the local government and contains sensitive information on individuals including gender, age and self-reported

## Abstract

### Background

Degenerative compressive myelopathy (DCM) is caused by cervical cord compression. The relationship between the magnitude and clinical findings of cervical cord compression has been described in the literature, but the details remain unclear. This study aimed to clarify the relationship between the magnitude and clinical symptoms of cervical cord compression in community-dwelling residents.

### Methods

The present study included 532 subjects. The subjective symptoms and the objective findings of one board-certified spine surgeon were assessed. The subjective symptoms were upper extremity pain and numbness, clumsy hand, fall in the past 1 year, and subjective gait disturbance. The objective findings were: Hoffmann, Trömner, and Wartenberg signs; Babinski's and Chaddock's signs; hyperreflexia of the patellar tendon and Achilles tendon reflexes; ankle clonus; Romberg and modified Romberg tests; grip and release test; finger escape sign; and grip strength. Using midsagittal T2-weighted magnetic resonance imaging, the anterior–posterior (AP) diameters (mm) of the spinal cord at the C2 midvertebral body level ($D_{C2}$) and at each intervertebral disc level from C2/3 to C7/T1 ($D_{C2/3-C7/T1}$) were measured. The spinal cord compression ratio (R) for each intervertebral disc level was defined and calculated as $D_{C2/3-C7/T1}$ divided by $D_{C2}$. The lowest R (LR) along C2/3 to C7/T1 of each individual was divided into 3 grades by the tertile method. The relationship between LR and clinical symptoms was investigated by trend analysis.

### Results

The prevalence of subjective gait disturbance increased significantly with the severity of spinal cord compression (p = 0.002812), whereas the other clinical symptoms were not significantly related with the severity of spinal cord compression.

data, and sharing these data openly is prohibited by the contract with the local government and Fukushima Medical University Ethics Committee. The authors did not have any special privileges to access the data and others would be able to access these data in the same manner as the authors. Data requests may be sent to Fukushima Medical University Ethics Committee (rs@fmu.ac.jp).

**Funding:** This study was supported by a grant from the Fukushima Society for the Promotion of Medicine and a grant from the Fukushima Prefectural Hospitals.

**Competing interests:** NO authors have competing interests.

## Conclusions

The magnitude of cervical cord compression had no relationship with any of the neurologic findings. However, subjective gait disturbance might be a better indicator of the possibility of early stage cervical cord compression.

## Introduction

Degenerative compressive myelopathy (DCM) can be caused by mechanical [1–5] and dynamic [6–9] compression of the cervical spinal cord, and has a variety of clinical presentations, including subjective symptoms and objective findings [10–15]. Disabilities secondary to the subjective symptoms of cervical myelopathy appear either in the upper extremities, lower extremities, or both. For example, clumsiness of the hand and fingers, and inability to grip, and dysesthetic pain are well-known characteristics of myelopathy hands [12]. In the lower extremity, unsteady gait, difficulty in going down and/or up stairs, and spontaneous muscle cramping may be present. Objective findings include long tract signs [12,14], which are brought about by the failure of the white matter [12,15] of the spinal cord conduction pathway. These signs comprise symptoms pertaining to the pyramidal tract, posterior column, and spinothalamic tract, and they include: clumsy hands; spastic paralysis; gait disturbance, including motor impairment of the lower extremities; hyperreflexia of the lower extremities; and the presence of Babinski's sign [12,14].

To date, numerous studies have described the relationship of severe cervical cord compression with the clinical symptoms of atrophy of the extrinsic and intrinsic hand muscles, clawing of the fingers, Hoffmann's sign, clumsy hands, and so on [16–25]. On the other hand, it is known that there are asymptomatic cervical cord compressions of the image. However, the initial symptom by the cervical cord compression is not clear. In the first place, it is not clear how much cervical cord compression causes clinical symptoms. The purpose of this study was to clarify the relationship between the magnitude and the clinical symptoms (i.e., subjective symptoms and objective findings) of single-level cord compression of the cervical spine, as evaluated by magnetic resonance imaging (MRI), in community-dwelling residents.

## Materials and methods

This study was approved by the ethics committee of Fukushima Medical University (No. 1880).

### Study design and subjects

In May, August, and November of 2005, in the annual checkups conducted by local governments for 3236 applicants (1326 men, 1910 women; age range, 19–94 years; average age, 65.5 years) of Tadami Town, Ina Village, and Tateiwa Village in mountainous areas of Fukushima Prefecture, Japan (Table 1), 582 people provided written, informed consent to undergo MRI, medical interviews, and physical examinations as a cervical spine medical examination by one board-certified spine surgeon (KO) in each place. When they were recruited, those who underwent cervical spinal cord surgery were excluded. All participants were self-sufficient; they lived in their own houses without the need for supplemental care and walked independently with or without support with a cane or a walker [26–28]. After a medical interview, neurological

**Table 1. Characteristics of annual checkup applicants and subjects of this study.**

| Annual checkup applicants Characteristic | Total (n = 3,236) | Male (n = 1,326) | Female (n = 1,910) |
|---|---|---|---|
| Age (y), mean (SD) | 65.5 (13.1) | 65.7 (13.3) | 65.3 (12.9) |
| Age range (%) | | | |
| ≤39 y | 170 (5.3) | 69 (5.2) | 101 (5.3) |
| 40–49 y | 247 (7.6) | 105 (7.9) | 142 (7.4) |
| 50–59 y | 493 (15.2) | 200 (15.1) | 293 (15.3) |
| 60–69 y | 849 (26.2) | 321 (24.2) | 528 (27.6) |
| 70–79 y | 1119 (34.6) | 473 (35.7) | 646 (33.8) |
| ≥80 y | 358 (11.1) | 158 (11.9) | 200 (10.5) |
| Subjects of this study Characteristic | Total (n = 532) | Male (n = 163) | Female (n = 369) |
| Age (y), mean (SD) | 64.2 (12.3) | 64.5 (12.1) | 64.1 (12.4) |
| Age range (%) | | | |
| ≤39 y | 25 (4.7) | 4 (2.5) | 21 (5.7) |
| 40–49 y | 42 (7.9) | 15 (9.2) | 27 (7.3) |
| 50–59 y | 106 (19.9) | 43 (26.4) | 63 (17.0) |
| 60–69 y | 143 (26.9) | 29 (17.8) | 114 (30.9) |
| 70–79 y | 181 (34.0) | 60 (36.8) | 121 (32.8) |
| ≥80 y | 35 (6.6) | 12 (7.4) | 23 (6.2) |

examination, and MRI, subjects with visual impairment, dementia, brain surgery, fracture of the lower extremities, and poor quality MRI were excluded. Finally, 532 subjects (163 men, 369 women; age range, 25–93 years and average age, 64.2 years) were available for analysis in this study (Table 1). More women than men participated in this study, and the most common age group was the 70s, with few in their 40s.

## Subjective symptoms

All subjective symptoms were determined from interviews conducted by KO and included upper extremity pain and numbness, clumsy hand, gait disturbance, and fall in the past 1 year. Clumsy hand was judged as positive when there was subjective impairment in at least 1 of 3 hand and finger actions, such as using chopsticks, writing, and fastening buttons. The number of fall episodes in the past 1 year was classified as 0, 1–2, or ≥3 times. In this study, 2 patterns based on the number of fall episodes were used for statistical analysis; these included pattern 1 (≥1 time) and pattern 2 (≥3 times) (Table 2). Gait disturbance was evaluated according to the lower extremity dysfunction score of the Japanese Orthopaedic Association (JOA) scoring system for cervical myelopathy (17–2) [29]. Gait disturbance was considered present when the lower extremity score was <3 points (Table 3).

**Table 2. Patterns of falls, romberg test, and modified romberg test.**

| Fall down | Pattern | | Romberg test | Pattern | | Modified Romberg test | Pattern | |
|---|---|---|---|---|---|---|---|---|
| | 1 | 2 | | 1 | 2 | | 1 | 2 |
| 0 times | negative | negative | (-) | negative | negative | (-) | negative | negative |
| 1–2 | positive | | (±) | positive | | (±) | positive | |
| ≥3 | | positive | (+) | | positive | (+) | | positive |

**Table 3. Assessment of gait disturbance.**

| | Gait disturbance by JOA score (17–2) | Gait disturbance |
|---|---|---|
| **4 points** | Normal | negative |
| **3** | Capable of fast walking but clumsy | |
| **2.5** | Walks independently when going up stairs but needs support when going down stairs | positive |
| **2** | Walks independently on a level but needs support on stairs | |
| **1.5** | Able to walk without a support but with a clumsy gait | |
| **1** | Unable to walk on a level without a cane or other support | |
| **0.5** | Able to stand up but unable to walk | |
| **0** | Unable to stand up and walk by any means | |

In this study, gait disturbance was defined by the JOA scoring system.

JOA: Japanese Orthopaedic Association.

## Objective findings

One experienced spine surgeon (KO) performed the neurologic examinations to evaluate the finger flexion reflexes (i.e., Hoffmann's sign [22], Trömner's sign [30], and Wartenberg's sign [31]); patellar tendon reflex (PTR); Achilles tendon reflex (ATR); ankle clonus [38]; and the pathological reflexes (i.e., Babinski's sign [32] and Chaddock's sign [33]). The finger flexion reflex was considered positive when flexion of the thumb was observed. PTR and ATR were assessed according to the National Institute of Neurological Disorders and Stroke Scale Myotatic Reflex Scale. Scale 4 was judged as hyperreflexia and represented an enhanced and more than normal reflex; it included clonus, if present, which can be optionally noted in the additional verbal description of the reflex [34,35]. The pathological reflexes were tested by stroking the lateral border of the sole the foot (Babinski's sign) or the lateral malleolar area (Chaddock's sign) with a blunt object and were considered present when dorsiflexion of the hallux in the proximal to distal direction was observed.

The Romberg test [30] and modified Romberg test [36,37] were performed with the eyes closed for more than 30 seconds while standing erect with feet together and on a straight line, respectively. The findings were classified into 3 categories, including (−) for stable, (±) for swaying but able to maintain a standing position, and (+) for impossible to maintain a standing position. In this study, 2 patterns were used for statistical analysis, as follows: pattern 1, when both (±) and (+) were positive; and pattern 2, when only (+) was positive (Table 2).

The finger escape sign (FES), which reflected motor dysfunction, was classified as grade 0–4 [15,38]. In this study, grade ≥1 was regarded as positive (Table 4). FES was assessed as positive if either the left or right hand was graded as >1. The grip and release test was conducted on

**Table 4. Assessment of FES.**

| Grade | Fingers | Deficiency | Assessment of FES |
|---|---|---|---|
| **0** | All | None | Negative |
| **1** | Little | Unable to hold adduction | Positive |
| **2** | Little or little and ring | Unable to assume adduction | |
| **3** | Little and ring | Unable to assume adduction or full extension | |
| **4** | Little, ring, and middle | Unable to assume adduction or full extension | |

FES: Finger escape sign.

FES was considered positive if at least 1 side was grade >1.

**Table 5. Cutoff value for the grip and release test [41] and grip strength [42].**

| Age (y) | 30–39 | | 40–49 | | 50–59 | | 60–69 | | 70–79 | | 80–89 | |
|---|---|---|---|---|---|---|---|---|---|---|---|---|
| Cutoff value for the grip and release test (times) | 21 | | 19 | | 19 | | 17 | | 14 | | 13 | |
| Cutoff value for grip strength (kg) | M | F | M | F | M | F | M | F | M | F | M | F |
| | 36 | 25 | 36 | 25 | 36 | 25 | 30 | 17 | 27 | 15 | 21 | 10 |

M: Male, F: Female.

The grip and release test was considered positive if at least 1 side was less than the cutoff value. Grip strength was considered positive if at least 1 side was less than the cutoff value.

the left and right hands. The subject was asked to grip and release the fingers (i.e., full finger flexion and extension) as rapidly as possible, and the number of movement cycles completed within 10 seconds was counted [15,39,40]. Grip strength of the left and right hands was assessed. Using the preliminary cutoff values reported in our previous study [41,42], the results of the grip and release test and grip strength were classified into 2 groups, including normal and impaired (positive) (Table 5). The grip and release test and grip strength were assessed as positive if one of the values in the left or right hand was less than the cutoff value.

## Magnetic resonance imaging

Midsagittal T2-weighted images were obtained using two MRI machines. All images were measured using a workstation (ZioCube, Mita, Minato-ku, Tokyo, Japan) at Fukushima Medical University (Fukushima City, Fukushima Prefecture) by one orthopedic surgeon (TH) who was blinded to the clinical information.

## Assessment of the degree of cervical cord compression

The anterior–posterior (AP) diameters (mm) of the spinal cord at the C2 midvertebral body level ($D_{C2}$) and at each intervertebral disc level from C2/3 to C7/T1 ($D_{C2/3-C7/T1}$) were measured using midsagittal T2-weighted images. There was no spinal cord compression in the C2 vertebral body level in all subjects. In the literature, there were individual differences in cervical cord size [43,44]. Because it was necessary to standardize the AP diameter of the spinal cord, the spinal cord compression ratio (R) was calculated, as shown in Fig 1.

Intra-observer and inter-observer reliabilities were calculated before the study results were analyzed. To evaluate intra-observer reliability, 30 MRIs of the cervical spine were randomly selected, and 180 AP diameters of the spinal cord (from C2-3 to C7-T1 of each) were measured three times by one observer (TH) every two weeks. Furthermore, to evaluate inter-observer reliability, other 30 MRIs were measured by two other orthopedic surgeons. In the measurement of AP diameter, intra-observer reliability was ρ = 0.73, and inter-observer reliability was ρ = 0.82. The intra-observer and inter-observer reliabilities were considered acceptable. Finally, all measurements were performed by TH and these measurements were adopted in this study.

In this study, the lowest R (LR) along the C2/3 to C7/T1 of each individual was classified into 3 grades by the tertile method (G1, G2, and G3) to assess single-level cord compression in the cervical spine.

## Statistical analysis

The distributions of age and sex in each grade were compared by the Jonckheere–Terpstra trend test. One-way analysis of variance was used to evaluate the differences in the average age

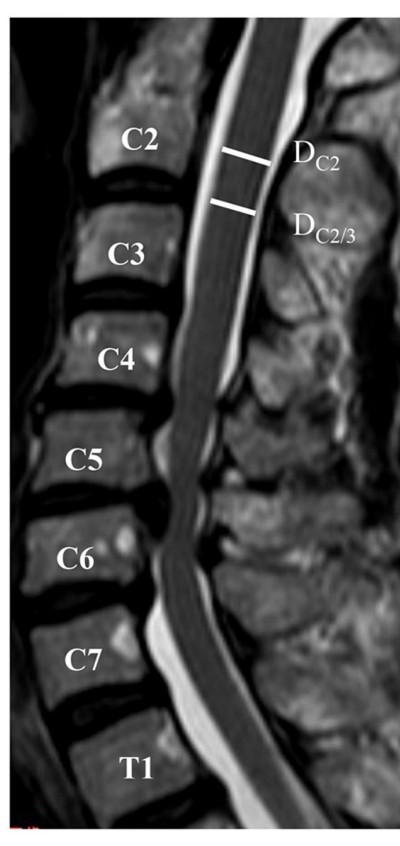

$D_{C2}$: AP diameter of the spinal cord in the C2 midvertebral body level (mm)

D: AP diameter of the spinal cord in the intervertebral disc level (mm) (C2/3-C7/T1)

$$R : \text{Spinal cord compression ratio} = \frac{D}{D_{C2}}$$

**Fig 1. Evaluation of spinal cord compression.** The spinal cord compression ratio (R) is used to evaluate the magnitude of cord compression. $D_{C2}$: AP diameter of the spinal cord in the C2 midvertebral body level (mm). D: AP diameter of the spinal cord in the intervertebral disc level (mm) (C2/3-C7/T1). AP: Anteroposterior.

among the 3 grades. The tendency for the prevalence of the clinical symptoms in each grade was evaluated by the Cochran–Armitage trend analysis. Data analyses were performed using IBM SPSS Statistics (ver. 24, SPSS Inc., Chicago, IL, USA) and R (version 3.4.3, Development Core Team, 2017). A p value of <0.05 was considered significant.

## Results

The distribution of LR is shown in Fig 2. The LR along C2/3 to C7/T1 of each individual ranged from 0.308 to 1.11; the 1st tertile was 0.71622 and the 2nd tertile was 0.78082. Based on these results, LR was divided into 3 grades, including G1 (LR >0.78082), G2 (0.78082 ≥ LR > 0.71622), and G3 (LR≤0.71622) to reflect the increase in the severity of cervical cord compression. The results for age and sex, subjective symptoms, and objective findings in the 3 grades are shown in Table 6. The severity of cord compression tended to increase with older age, but this was not significant. The sex distribution was almost the same among the 3 grades. All subjective symptoms, except gait disturbance, were not significantly related to the severity of spinal cord compression. Only the prevalence of gait disturbance increased significantly with the severity of spinal cord compression (p = 0.002812) [G1 (26 subjects, 15.0%), G2 (25 subjects, 14.2%), and G3 (50 subjects, 27.3%)]. On the other hand, all objective findings were not significantly related to the severity of spinal cord compression.

The prevalence of gait disturbance increased significantly with the severity of spinal cord compression.

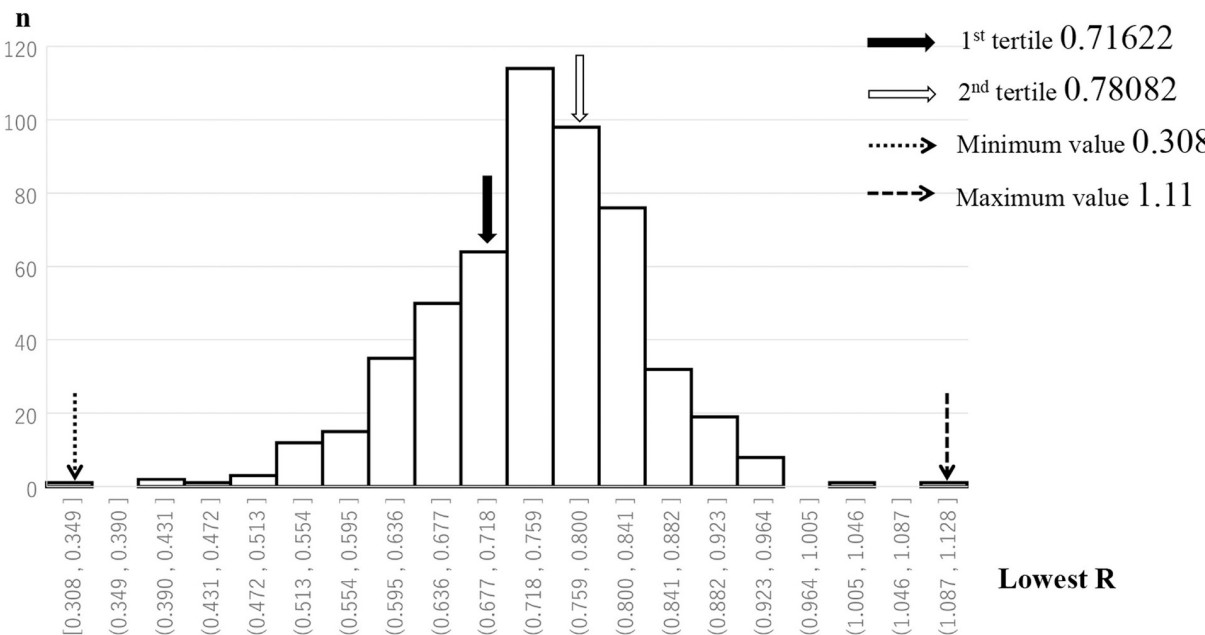

**Fig 2. Range of the lowest R (LR).** The LR has a minimum value of 0.308 and a maximum value of 1.11. The 1st tertile is 0.71622, and the 2nd tertile is 0.78082.

## Discussion

Severe spinal cord compression is widely known to induce clinical symptoms [1,6]. In an autopsy study, an AP diameter of <40% of the normal cervical spinal cord diameter was reported to show severe degenerative changes in the white matter [17]. Similarly, in the clinical and hospital settings, the magnitude of cervical spinal cord compression was reported to be related to clinical symptoms, JOA score, and the postoperative recovery rate of the JOA score [22,45–52]. However, cervical cord compression is not always directly related to the presence of subjective and objective symptoms [53–59]. Boden et al reported asymptomatic cervical disc degeneration in 25% of subjects <40 years old and in almost 60% of subjects >40 years old [53]. Matsumoto et al reported that posterior protrusion and even compression of the spinal cord were not rare in asymptomatic subjects >40 years of age [60]. Moreover, Teresi et al found asymptomatic cervical disc protrusions in 20% of patients aged 45 to 54 years [55]. Based on these reports, cervical cord compression does not always induce symptoms. Therefore, the relationship between the magnitude and the onset or degree of symptoms of cervical cord compression remains unclear.

In the present study, the magnitude of cervical cord compression was evaluated as the spinal cord compression ratio on MRI. Several measurement methods for the magnitude of cervical cord compression have been reported, including the AP diameter ratio of the compressed spinal cord to the spinal canal and the AP diameter ratio of the normal (not compressed) spinal cord to the transverse area [49,60–64]. Kameyama et al reported that the transverse area of the C7 segment varied from 33.3 to 74.0 mm$^2$ in 152 cadaveric specimens [43]. This fact suggested that the individual size of the spinal cord varies widely. Therefore, the use of a relative value might be suitable for comparison of individual data and for categorization of the severity of spinal cord compression into 3 grades.

According to the results of the present study, the magnitude of cervical cord compression was related to subjective gait disturbance but not to objective findings. The Wakayama Study

**Table 6. Results for age, sex and the prevalence of subjective symptoms and objective findings in each grade.**

| | | G1 | G2 | G3 | P |
|---|---|---|---|---|---|
| the lowest R (LR) | | LR>0.78082 | 0.78082≥LR>0.71622 | 0.71622≥LR | |
| n | | 173 | 176 | 183 | |
| Age (years) | <40 | 7 | 11 | 7 | 0.075 |
| | 40–49 | 20 | 11 | 11 | |
| | 50–59 | 32 | 41 | 33 | |
| | 60–69 | 48 | 44 | 51 | |
| | 70–79 | 58 | 60 | 63 | |
| | ≥80 | 8 | 9 | 18 | |
| | Average ± S.D. | 63.4±12.4 | 63.3±12.4 | 66.0±12.0 | 0.059 |
| Sex | Male | 55 | 55 | 53 | 0.56 |
| | Female | 118 | 121 | 130 | |
| Subjective symptoms | Upper extremity pain | 23 | 20 | 34 | 0.15 |
| | Upper extremity numbness | 40 | 23 | 46 | 0.6038 |
| | Clumsy hand | 11 | 12 | 16 | 0.3855 |
| | *Gait disturbance | 26 | 25 | 50 | 0.002812 |
| | Fall down 1 (≥1) | 40 | 32 | 47 | 0.5445 |
| | Fall down 2 (≥3) | 7 | 12 | 10 | 0.5665 |
| Objective findings | Hoffmann's reflex | 17 | 14 | 17 | 0.8674 |
| | Trömner reflex | 9 | 5 | 11 | 0.5369 |
| | Wartenberg reflex | 33 | 26 | 33 | 0.8085 |
| | Hyperreflexia of the PTR | 1 | 0 | 0 | 0.2124 |
| | Hyperreflexia of the ATR | 5 | 4 | 7 | 0.5979 |
| | Ankle clonus | 6 | 5 | 11 | 0.222 |
| | Babinski reflex | 0 | 0 | 1 | 0.2298 |
| | Chaddock reflex | 0 | 0 | 0 | NA |
| | Grip and release test | 17 | 21 | 26 | 0.2039 |
| | Grip strength | 36 | 30 | 47 | 0.25 |
| | Finger escape sign | 18 | 23 | 20 | 0.8871 |
| | Romberg test 1 | 140 | 137 | 157 | 0.2266 |
| | Romberg test 2 | 2 | 1 | 3 | 0.656 |
| | Modified Romberg test 1 (± or +) | 158 | 164 | 170 | 0.5801 |
| | Modified Romberg test 2 (+) | 104 | 101 | 120 | 0.2821 |

S.D.: Standard deviation.

PTR: Patellar tendon reflex.

ATR: Achilles tendon reflex.

NA: Not available.

[65] of community-dwelling residents reported similar results and showed that cervical cord compression was associated with physical performance (i.e., grip and release test, 6-m walking time at a maximal pace, step length at a usual and maximal pace, and chair stand time), but not with myelopathy signs (i.e., hyperreflexia of the PTR, Hoffmann's sign, and Babinski's sign). That study finally concluded that cervical cord compression correlated with physical performance, and that impairment of physical performance could be detected in the early stage of the disease before the appearance of objective myelopathy signs. Moreover, other

studies suggested that gait disturbance was one of the early symptoms of cervical compressive myelopathy [66–72]. In the literature, it is not clear how much cervical cord pressure results in physical symptoms. In contrast, it was clear in the present study that the prevalence of subjective gait disturbance increased if LR was less than 0.71622. In other words, the magnitude of cervical cord compression was successfully quantized. This is considered to be the most valuable point in this study.

As suggested by the results of the present study, subjective gait disturbance based on the JOA score, compared with physical performance, may be a better indicator of the possibility of cervical cord compression, which is the early stage of DCM. This implies that clinicians should keep in mind the possibility of cervical cord compression or early stage DCM in patients with subjective complaints of gait disturbance before the occurrence of any neurologic deterioration.

There were several limitations [26–28] in this study. First, comorbidities, such as osteoarthritis of the hip and knee, lumbar spinal stenosis, and cerebrovascular disease, including Parkinson syndrome, which can influence gait ability, were not excluded. Second, only one experienced spine surgeon performed the neurologic examinations, and the reliability of each procedure was not assessed. Third, there was no evaluation of cervical radiculopathy and peripheral neuropathy, including carpal tunnel syndrome and cubital tunnel syndrome. Fourth, the research location was in a rural and mountainous area; therefore, the data may not be extrapolated completely to the typical Japanese population. Fifth, we did not consider the sample size and do a power calculation before starting this study. Because we could not assume the number of persons who underwent MRI. As a result, there was not representative of the actual age demographics in these places, especially concerning the population of people under the age of 40 years. It is necessary to consider whether the results of this study can be applied to young people. Finally, all of the participants in this study were volunteers and, as such, there could have been an inevitable sample bias. Although this study had limitations, it clarified the relationship between the magnitude of cervical cord compression and subjective gait disturbance in community-dwelling residents.

## Conclusion

In community-dwelling residents, the magnitude of cervical cord compression was related to the presence of subjective gait disturbance, but not objective findings. Therefore, subjective gait disturbance might be a good indicator of the possibility of early stage DCM.

## Supporting information

**S1 Table. Detail of MRI.**
(DOCX)

**S1 File. Doctor interview sheet in English.**
(DOC)

**S2 File. Doctor interview sheet in Japanese.**
(DOC)

## Acknowledgments

The authors would like to thank Dr. Akira Onda, Dr. Kazuya Yamauchi, Dr. Yoshiaki Takeyachi, Dr. Ichiro Takahashi, Dr. Hisayoshi Tachihara, and Dr. Bunji Takayama for participating in data collection. The authors would also like to thank five public health nurses, Ms. Nobuko

Fujita, Ms. Nakako Hoshi, Ms. Misako Hoshi, Ms. Naoko Imada, and Ms. Seiko Kanno, for their support in carrying out this study.

## Author Contributions

**Conceptualization:** Toru Hirai, Koji Otani.

**Data curation:** Koji Otani.

**Formal analysis:** Toru Hirai.

**Investigation:** Toru Hirai, Koji Otani.

**Methodology:** Koji Otani.

**Project administration:** Koji Otani.

**Supervision:** Koji Otani, Miho Sekiguchi, Shin-ichi Kikuchi, Shin-ichi Konno.

**Writing – original draft:** Toru Hirai.

**Writing – review & editing:** Koji Otani, Miho Sekiguchi, Shin-ichi Konno.

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
