## [Decision Letter · Decision Letter 0]

5 Feb 2021

PONE-D-20-32795

Epidemiological study on cervical cord compression and its clinical symptoms in community-dwelling residents

PLOS ONE

Dear Dr. Hirai,

Thank you for submitting your manuscript to PLOS ONE. After careful consideration, we feel that it has merit but does not fully meet PLOS ONE’s publication criteria as it currently stands. Therefore, we invite you to submit a revised version of the manuscript that addresses the points raised during the review process.

We look forward to receiving your revised manuscript.

Kind regards,

Gregory W.J. Hawryluk, MD, PhD, FRCSC

Academic Editor

PLOS ONE

Journal Requirements:

2. Please note that PLOS does not permit references to “data not shown.” Authors should provide the relevant data within the manuscript, the Supporting Information files, or in a public repository. If the data are not a core part of the research study being presented, we ask that authors remove any references to these data.

3. In your Methods section, please provide additional information about the participant recruitment method and the demographic details of your participants. Please ensure you have provided sufficient details to replicate the analyses such as: a) the recruitment date range (month and year), b) a description of any inclusion/exclusion criteria that were applied to participant recruitment, c) a statement as to whether your sample can be considered representative of a larger population, d a description of how participants were recruited, and e) descriptions of where the research took place.

4. Please provide a sample size and power calculation in the Methods, or discuss the reasons for not performing one before study initiation.

5. Please include additional information regarding the survey or questionnaire used in the study and ensure that you have provided sufficient details that others could replicate the analyses. For instance, if you developed a questionnaire as part of this study and it is not under a copyright more restrictive than CC-BY, please include a copy, in both the original language and English, as Supporting Information.

6.Thank you for stating the following in the Source of funding Section of your manuscript:

"study was supported by a grant from the Fukushima Society for the

Promotion of Medicine and a grant from the Fukushima Prefectural Hospitals. These

grants were mainly used for the cost of the MRI, research assistance, and printing and

posting the questionnaire."

 "unfunded studies"

7. Please amend the manuscript submission data (via Edit Submission) to include author Koji Otani, MD, DMSc, Miho Sekiguchi, MD, PhD, Shin-ichi Kikuchi, MD, PhD, and Shin-ichi Konno.

8. Your ethics statement should only appear in the Methods section of your manuscript. If your ethics statement is written in any section besides the Methods, please move it to the Methods section and delete it from any other section. Please ensure that your ethics statement is included in your manuscript, as the ethics statement entered into the online submission form will not be published alongside your manuscript.

9. Please include your tables as part of your main manuscript and remove the individual files. Please note that supplementary tables (should remain/ be uploaded) as separate "supporting information" files.

Additional Editor Comments:

There seems to have been a problem with the ability of the reviewers to access the manuscript for review. I am cognizant of the lengthy wait to get this manuscript reviewed. I think it would be fair to have the authors address the revisions suggested by Reviewer #2 and to re-engage reviewer #1 at that time as opposed to now taking on an anticipated lengthy delay to get feedback from another reviewer.

Reviewers' comments:

Reviewer's Responses to Questions

**Comments to the Author**

1. Is the manuscript technically sound, and do the data support the conclusions?

Reviewer #1: No

Reviewer #2: Yes

2. Has the statistical analysis been performed appropriately and rigorously? 

Reviewer #1: No

Reviewer #2: Yes

3. Have the authors made all data underlying the findings in their manuscript fully available?

Reviewer #1: Yes

Reviewer #2: Yes

4. Is the manuscript presented in an intelligible fashion and written in standard English?

Reviewer #1: No

Reviewer #2: No

5. Review Comments to the Author

Reviewer #1: The authors have submitted what appears to be figures and graphs but no actual manuscript body. It is impossible to evaluate the quality of data without this. Hopefully this material will be provided at some point.

Reviewer #2: This is an interesting and relevant topic. The relationship between cord compression and clinical signs and symptoms is of increasing importance as this problem becomes more prevalent in the aging population. However there are some concerns:

1) It is not clear how novel this study is. Many of the references are somewhat dated. The introduction refers to "numerous studies" and that the "relationship remains unclear". This doesn't really demonstrate why this study is any different than any of the other "numerous studies" besides adding incrementally to the literature.

2) While the English and grammar are not terrible, there are areas where the grammar and use of non-scientific language could be improved.

3) It wasn't clear to me whether the examining surgeon was blinded to the MRI results at the time of examination.

4) There is a missed opportunity here to also examine cumulative compression. The data are all there, the statistics just need to be run.

5) For the AP diameter measurement, it is not clear how many radiologists were utilized to obtain the intra and inter-observer reliability. Then this was deemed "acceptable" and only one radiologist's measurements were utilized. It is unclear whether this reliability was pre-determined or was determined using the data for this study. If more than one radiologist read for this study, why not average their readings for more accuracy and input? More information could be provided here.

Overall, there are some interesting findings here, but needs some revision.

6. PLOS authors have the option to publish the peer review history of their article (what does this mean?). If published, this will include your full peer review and any attached files.

Reviewer #1: No

Reviewer #2: **Yes: **Ann M. Parr

---

## [Author Response · Author response to Decision Letter 0]

27 May 2021

Response to journal requirements

Response: Thank you very much for your comments. We have made the necessary changes.

2. Please note that PLOS does not permit references to “data not shown.” Authors should provide the relevant data within the manuscript, the Supporting Information files, or in a public repository. If the data are not a core part of the research study being presented, we ask that authors remove any references to these data.

Response: We have deleted “data not shown”.

Assessment of the degree of cervical cord compression

P11, Line 181. In the measurement of AP diameter, intra-observer reliability was ρ = 0.73, and inter-observer reliability was ρ = 0.82. [data not shown].

3. In your Methods section, please provide additional information about the participant recruitment method and the demographic details of your participants. Please ensure you have provided sufficient details to replicate the analyses such as: 

a) the recruitment date range (month and year), 

b) a description of any inclusion/exclusion criteria that were applied to participant recruitment, 

c) a statement as to whether your sample can be considered representative of a larger population, 

d) a description of how participants were recruited, and 

e) descriptions of where the research took place.

Response: We have provided the information requested.

a) This study was performed in March, August, and November of 2005.

Materials and Methods 

P4, line 77. In May, August, and November of 2005, ・・・

b) In the annual checkup for 3236 applicants by local governments, subjects who underwent cervical MRI were recruited. When they were recruited, those who underwent cervical spinal cord surgery were excluded. After a medical interview, neurological examination, and MRI, subjects with visual impairment, dementia, brain surgery, fracture of the lower extremities, and poor quality MRI were excluded.

Materials and Methods 

P5, Line 83. When they were recruited, those who underwent cervical spinal cord surgery were excluded. 

P5, Line 86. After a medical interview, neurological examination, and MRI, subjects with visual impairment, dementia, brain surgery, fracture of the lower extremities, and poor quality MRI were excluded.

.

c) The number of subjects varied by age group. Therefore, the following was added regarding generalizability.

Discussion 

P18, line 286.・・・there was not representative of the actual age demographics in these places, especially concerning the population of people under the age of 40 years. It is necessary to consider whether the results of this study can be applied to young people. 

d) e) In the annual checkup of 3236 applicants, 582 people provided written, informed consent to undergo MRI, medical interview, and physical examination as a cervical spine medical examination by one board-certified spine surgeon (KO) at Tadami Town, Ina Village, and Tateiwa Village in Fukushima Prefecture. AP diameters of the cervical spinal cord were then measured using a workstation at Fukushima Medical University (Fukushima City, Fukushima Prefecture).

Materials and Methods 

P4, line 77. 

・・・in the annual checkups conducted by local governments for 3236 applicants・・・, 582 people provided written, informed consent to undergo MRI, medical interviews, and physical examinations as a cervical spine medical examination by one board-certified spine surgeon (KO) in each place.

P11, line 163． All images were measured using a workstation (ZioCube, Mita, Minato-ku, Tokyo, Japan) at Fukushima Medical University (Fukushima City, Fukushima Prefecture) by one orthopedic surgeon (TH) who was blinded to the clinical information.

4. Please provide a sample size and power calculation in the Methods, or discuss the reasons for not performing one before study initiation.

Response: We did not consider the sample size and do a power calculation before starting this study because we could not assume the number of persons who underwent MRI. A total of 582 subjects in this study were recruited from among those who underwent the annual checkup (3236 applicants). The age and sex distributions were approximately similar in the two groups (Table 1). Therefore, it was possible to think that the subjects were representative of the 3236 applicants. However, there is not representative of the actual age demographics in Tadami Town, Ina Village, and Tateiwa Village, especially concerning the population of people under the age of 40 years. It is necessary to consider whether the results of this study can be applied to young people. This point is described as one of the limitations.

Discussion 

P18, line 284．Fifth, we did not consider the sample size and do a power calculation before starting this study. Because we could not assume the number of persons who underwent MRI. As a result, there was not representative of the actual age demographics in these places, especially concerning the population of people under the age of 40 years. It is necessary to consider whether the results of this study can be applied to young people.

5. Please include additional information regarding the survey or questionnaire used in the study and ensure that you have provided sufficient details that others could replicate the analyses. For instance, if you developed a questionnaire as part of this study and it is not under a copyright more restrictive than CC-BY, please include a copy, in both the original language and English, as Supporting Information.

Response: In this study, subjective symptoms were investigated based on an interview sheet. We have provided the interview sheet in both the original language and English.

6.Thank you for stating the following in the Source of funding Section of your manuscript:

"study was supported by a grant from the Fukushima Society for the Promotion of Medicine and a grant from the Fukushima Prefectural Hospitals. These

grants were mainly used for the cost of the MRI, research assistance, and printing and

posting the questionnaire."

 "unfunded studies"

Response: We have removed funding-related text. We have also corrected the funding statement according to your comments.

7. Please amend the manuscript submission data (via Edit Submission) to include author Koji Otani, MD, DMSc, Miho Sekiguchi, MD, PhD, Shin-ichi Kikuchi, MD, PhD, and Shin-ichi Konno.

Response: We have amended the manuscript submission data

8. Your ethics statement should only appear in the Methods section of your manuscript. If your ethics statement is written in any section besides the Methods, please move it to the Methods section and delete it from any other section. Please ensure that your ethics statement is included in your manuscript, as the ethics statement entered into the online submission form will not be published alongside your manuscript.

Response: The ethics statement has been moved to the Methods suggestion, as required.

Material and Methods

P4, line 74. This study was approved by the ethics committee of our university (No. 1880).

9. Please include your tables as part of your main manuscript and remove the individual files. Please note that supplementary tables (should remain/ be uploaded) as separate "supporting information" files.

Response: We have made the necessary changes.

Reviewer #1: The authors have submitted what appears to be figures and graphs but no actual manuscript body. It is impossible to evaluate the quality of data without this. Hopefully this material will be provided at some point.

Response: We have put the figures and graphs in the manuscript body.

Reviewer #2

1) It is not clear how novel this study is. Many of the references are somewhat dated. The introduction refers to "numerous studies" and that the "relationship remains unclear". This doesn't really demonstrate why this study is any different than any of the other "numerous studies" besides adding incrementally to the literature.

Response: Thank you very much for your comments. As you indicate, it is unclear in the manuscript why this study is any different than other studies. In the literature, it was not clear how much cervical cord pressure (degree of compression of the spinal cord) resulted in physical symptoms. Therefore, we tried to quantify the degree of compression and examined its relationship to symptoms. We believe this is clinically important and valuable.

Introduction 

P4, line 65 

On the other hand, it is known that there are asymptomatic cervical cord compressions of the image. However, the initial symptom by the cervical cord compression is not clear. In the first place, it is not clear how much cervical cord compression causes clinical symptoms.

2) While the English and grammar are not terrible, there are areas where the grammar and use of non-scientific language could be improved.

Response: This manuscript was checked by a professional native English-speaking medical editor associated with Forte Science Communications (Tokyo, Japan) again. We have included their certificate of English-language editing.

3) It wasn't clear to me whether the examining surgeon was blinded to the MRI results at the time of examination.

Response: Subjective symptoms and objective findings were evaluated by KO before MRI. On the other hand, AP diameter was measured by TH who was blinded to the information about subjective symptoms and objective findings. Finally, these data were collated when they were analyzed.

Material and Methods

P4, line80．・・・582 people provided written, informed consent to undergo MRI, medical interviews, and physical examinations as a cervical spine medical examination by one board-certified spine surgeon (KO) in each place.

P11, line 163． All images were measured using a workstation (ZioCube, Mita, Minato-ku, Tokyo, Japan) at Fukushima Medical University (Fukushima City, Fukushima Prefecture) by one orthopedic surgeon (TH) who was blinded to the clinical information.

4) There is a missed opportunity here to also examine cumulative compression. The data are all there, the statistics just need to be run.

Response: The purpose of this study was to clarify the relationship between the magnitude and the clinical symptoms of single-level cord compression at the maximal pressure part of the cervical spine. We would like to investigate cumulative compression in our next study.

5) For the AP diameter measurement, it is not clear how many radiologists were utilized to obtain the intra and inter-observer reliability. Then this was deemed "acceptable" and only one radiologist's measurements were utilized. It is unclear whether this reliability was pre-determined or was determined using the data for this study. If more than one radiologist read for this study, why not average their readings for more accuracy and input? More information could be provided here.

Response: Inter-observer and intra-observer reliabilities were calculated before the analysis. 30 subjects were randomly chosen. It was judged that the results for intra-observer reliability (ρ = 0.73) and inter-observer reliability (ρ = 0.82) were acceptable. Therefore the measurements obtained by TH were analyzed in this study. This is because if one doctor's measurements are statistically reliable, they are more efficient than the average of multiple doctors' measurements, we considered.

Material and Methods

P11, line 176. Intra-observer and inter-observer reliabilities were calculated before the study results were analyzed. To evaluate intra-observer reliability, 30 MRIs of the cervical spine were randomly selected, and 180 AP diameters of the spinal cord (from C2-3 to C7-T1 of each) were measured three times by one observer (TH) every two weeks. Furthermore, to evaluate inter-observer reliability, other 30 MRIs were measured by two other orthopedic surgeons. In the measurement of AP diameter, intra-observer reliability was ρ = 0.73, and inter-observer reliability was ρ = 0.82. The intra-observer and inter-observer reliabilities were considered acceptable. Finally, all measurements were performed by TH and these measurements were adopted in this study.

---

## [Decision Letter · Decision Letter 1]

16 Aug 2021

Epidemiological study on cervical cord compression and its clinical symptoms in community-dwelling residents

PONE-D-20-32795R1

Dear Dr. Otani,

We’re pleased to inform you that your manuscript has been judged scientifically suitable for publication and will be formally accepted for publication once it meets all outstanding technical requirements.

Kind regards,

Gregory W.J. Hawryluk, MD, PhD, FRCSC

Academic Editor

PLOS ONE

Additional Editor Comments (optional):

Thank you for addressing the reviewer critiques.

Reviewers' comments:

Reviewer's Responses to Questions

**Comments to the Author**

1. If the authors have adequately addressed your comments raised in a previous round of review and you feel that this manuscript is now acceptable for publication, you may indicate that here to bypass the “Comments to the Author” section, enter your conflict of interest statement in the “Confidential to Editor” section, and submit your "Accept" recommendation.

Reviewer #2: All comments have been addressed

2. Is the manuscript technically sound, and do the data support the conclusions?

Reviewer #2: Yes

3. Has the statistical analysis been performed appropriately and rigorously? 

Reviewer #2: Yes

4. Have the authors made all data underlying the findings in their manuscript fully available?

Reviewer #2: Yes

5. Is the manuscript presented in an intelligible fashion and written in standard English?

Reviewer #2: Yes

6. Review Comments to the Author

Reviewer #2: (No Response)

7. PLOS authors have the option to publish the peer review history of their article (what does this mean?). If published, this will include your full peer review and any attached files.

Reviewer #2: No

---

## [Editor Report · Acceptance letter]

19 Aug 2021

PONE-D-20-32795R1 

Epidemiological study of cervical cord compression and its clinical symptoms in community-dwelling residents 

Dear Dr. Otani:

I'm pleased to inform you that your manuscript has been deemed suitable for publication in PLOS ONE. Congratulations! Your manuscript is now with our production department. 

Kind regards, 

on behalf of

Dr. Gregory W.J. Hawryluk 

Academic Editor

PLOS ONE